# SCISplat: 3D Gaussian Splatting from a Snapshot Compressive Image

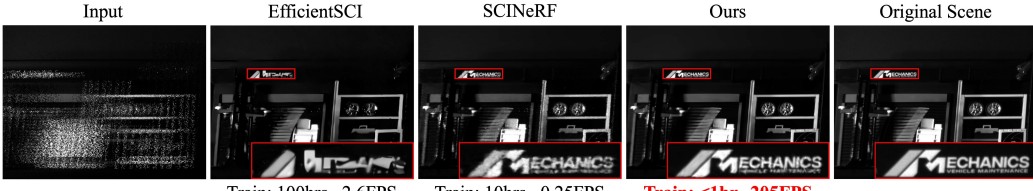

Figure 1: Given a single snapshot compressed image, our method can recover the underlying 3D scene representation. Leveraging the fast radiance field representation of 3D Gaussian Splatting, we can render high-quality images from a **single measurement** in **real-time**.

## ABSTRACT

In this paper, we investigate the potential of Snapshot Compressive Imaging (SCI) for efficiently recovering 3D scenes from a single temporally compressed image. SCI offers a cost-effective approach using a series of 2D masks to compress video data into a single image captured by 2D imaging sensors. However, traditional SCI reconstruction methods face challenges with generalization and maintaining multi-view consistency. Recent advances have introduced Neural Radiance Fields (NeRF) to estimate 3D scenes from SCI images, but NeRF's implicit representation struggles to capture fine details and support fast training and rendering. To address these issues, we propose SCISplat, a 3D Gaussian Splatting-based framework for decoding SCI images and achieving high-quality scene reconstruction from a single SCI image. First, we design an initialization protocol that robustly estimates the initial point cloud and camera poses from an SCI image, leveraging a learning-based Structure-from-Motion method. Second, we integrate the SCI image formation model into the 3D Gaussian training process and jointly optimize the Gaussians and camera poses to enhance reconstruction quality. Experiments demonstrate that SCISplat surpasses state-of-the-art methods, achieving a **2.3 dB** improvement in reconstruction quality and a **10×** faster training speed. Furthermore, results on real-world datasets show that our approach produces cleaner and sharper details, underscoring its practical value.

## 1 INTRODUCTION

Compressing 3D information of a scene in a single snapshot compressive image could vastly reduce the storage requirement and transmission bandwidth. For instance, a fast-moving car or quadrotor may capture the street view around with SCI instead of an expensive high-speed camera. A systematic method to effectively reconstruct the 3D environment from these snapshots is needed in this case. However, reconstructing the underlying 3D scene structure directly from a single snapshot compressive image has yet to be well-explored. A typical video SCI system (Yuan et al., 2021a) contains a hardware encoder and software decoder. The encoder captures the scene, and input images are optically modulated with temporally varying masks to form a single SCI measurement, which a 2D camera can record. Then, for the software decoding, the compressed measurement and corresponding modulation masks are fed into a reconstruction algorithm to recover the fully captured images of the scene. In recent years, many SCI reconstruction algorithms have been developed, ranging from traditional model-based methods (Yuan, 2016) to deep-learning-based methods (Yuan et al., 2020; 2021b; Wang et al., 2023). These deep-learning-based methods usually exceed traditional methods by a large margin in terms of reconstruction quality and speed. However, these

methods need to be pre-trained with an extensive collection of images for about 100 hours and do not generalize well for novel modulation masks, which means the model needs to be fine-tuned every time before performing the actual reconstruction. Recently, Li et al. (2024) proposed SCINeRF, which employs an implicit neural representation to explore underlying 3D scene structure from SCI measurements. SCINeRF is agonistic to different patterns of modulation masks. However, it suffers from a substantially long reconstruction time of about 10 hours for a single measurement and a slow inference speed of 0.25FPS. A recent advancement, 3D Gaussian Splatting (3DGS) (Kerbl et al., 2023), extends the implicit neural rendering to explicit 3D Gaussians. By projecting these optimized Gaussians onto the image plane, 3DGS enables real-time rendering, significantly improving both the efficiency and rendering quality compared to NeRF during training and inference.

To this end, we propose SCISplat, the first 3DGS-based approach to efficiently reconstruct explicit 3D scene structure from a single snapshot compressive image. However, naively applying 3DGS to a single SCI image is challenging. First, initializing 3D Gaussians with point cloud and camera poses using COLMAP (Schönberger & Frahm, 2016) from a single snapshot compressive image is impossible because of measurement noise and information loss caused by compression. Second, the original 3DGS requires multi-view images for supervision, while we only have one SCI image. To address these challenges, we first propose a novel initialization protocol. It first quickly decodes a sequence of degraded frames from the measurement that contains low-frequency cues of the 3D scene, then robustly estimates the initial point cloud and poses from degraded frames to kick off Gaussians training by leveraging a learning-based Structure-from-Motion (SfM) method, VG-GSfM (Wang et al., 2024a). After initialization, we jointly optimize Gaussians and camera poses to ensure higher reconstruction quality and trajectory accuracy. Finally, we apply a specifically designed loss function by incorporating the SCI image formation model with the 3DGS training procedure. It minimizes photometric loss between synthesized measurements from differentiable Gaussian rasterization and real SCI measurements. Moreover, to compensate for the ill-posedness of SCI image reconstruction, we optimize Gaussians with Monte Carlo Markov Chain (MCMC) strategy (Kheradmand et al., 2024) that effectively suppresses noise.

We conduct extensive experiments on synthetic and real datasets to evaluate our method properly. Regarding rendering quality, our method exceeds the SOTA SCINeRF method by **2.3 dB** on synthetic datasets and shows visually sharper reconstruction results on real datasets. As for efficiency, our method can consistently yield better results than SCINeRF with **820×** faster inference/rendering speed and **10×** faster training speed.

In summary, our contributions are listed as follows:

- We propose the first 3DGS-based SCI decoding method that can efficiently restore compressed images from a single snapshot compressive image and is also agnostic to modulation masks of different patterns.

- We propose an initialization protocol to derive point clouds and poses from a single snapshot compressive image robustly, which will benefit any downstream 3D task from SCI measurements.

- We show that our method surpasses the current state-of-the-art (SOTA) method in terms of reconstruction quality and training speed with extensive experiments on synthetic and real datasets.

## 2 RELATED WORK

**Video SCI Reconstruction.** Early methods for SCI image decoding primarily relied on regularized optimization-based approaches (Yang et al., 2020; Liao et al., 2014; Yuan, 2016; Liu et al., 2018). These techniques estimate compressed images by iteratively solving optimization problems with various regularizers, such as sparsity (Yang et al., 2020) and total variation (TV) (Yuan, 2016). Instead of gradient descent, most methods utilize the alternating direction method of multipliers (ADMM) (Boyd et al., 2011), which provides better results and adaptability across different systems. Notable approaches include decompress SCI (Liu et al., 2018) and GAP-TV (Yuan, 2016). However, these methods suffer from long runtimes and limited flexibility for high-resolution images. With the advent of deep learning, many recent SCI decoding methods have shifted to employing deep neural networks like U-net (Ronneberger et al., 2015) and GAN (Goodfellow et al., 2014). These

learning-based methods require extensive training data, often comprising thousands or even millions of synthetic SCI measurements and masks, which can be expensive. The networks are optimized using various losses, such as mean squared error (MSE), feature loss (Johnson et al., 2016), and GAN loss (Miao et al., 2019). Qiao et al. (2020) developed an end-to-end CNN using reconstruction loss to recover compressed images. Cheng et al. (2020) proposed a bidirectional recurrent neural network architecture to reconstruct temporal frames sequentially. RevSCI (Cheng et al., 2021) addressed the time and memory limitations of large-scale video SCI training by introducing a multi-group reversible 3D CNN architecture. ADMM-Net (Ma et al., 2019) modeled the decoding process as a tensor recovery problem from random linear measurements, interpreting the ADMM process as a deep neural network. MetaSCI (Wang et al., 2021) utilized a meta-modulated CNN to enhance the adaptability of the reconstruction network for large-scale data and novel masks. Plug-and-play fast and flexible denoising CNN (PnP-FFDNet) (Yuan et al., 2020) combined deep denoising networks with ADMM, enabling quick and flexible reconstruction. Later, Yuan et al. (2021b) developed a fast deep video denoising network (FastDVDNet), enhancing PnP-FFDNet's performance with the latest deep denoising techniques. Wang et al. introduced spatial-temporal transformers and EfficientSCI (Wang et al., 2023; Cao et al., 2024) to exploit spatial and temporal correlations within the image decoding process using Transformer (Vaswani et al., 2017) architecture. These methods require pre-training on synthetic datasets and thus may not generalize well to real datasets. Moreover, existing deep learning-based methods can only reconstruct 2D images corresponding to the masks, making it challenging to restore view-consistent images and estimate 3D scenes. Li et al. (2024) proposed SCINeRF, which recovers the underlying 3D scene from an SCI image by jointly optimizing camera poses and NeRF. However, due to implicit scene representation, SCINeRF suffers from high-frequency noises and low training/rendering speeds.

**Efficient Radiance Field Rendering.** NeRF (Mildenhall et al., 2021) revolutionized 3D scene representation using multi-layer perceptron (MLP), achieving impressive results in novel view synthesis. However, it struggles with long training times due to its costly volumetric rendering process. To address these challenges, various adaptations have emerged, such as grid-based methods like TensoRF (Chen et al., 2022), Plenoxels (Fridovich-Keil et al., 2022), and HexPlane (Cao & Johnson, 2023), and hash-based methods like InstantNGP (Müller et al., 2022). Despite these advancements, rendering speed remains a critical bottleneck. 3DGS (Kerbl et al., 2023) overcomes some of these challenges by enabling real-time rendering with quality comparable to state-of-the-art NeRF methods through an efficient explicit representation. However, 3DGS relies heavily on COLMAP (Schönberger & Frahm, 2016) and multi-view sharp images for generating high-quality point clouds and camera poses, which are sensitive to the quality of initial point clouds and camera parameters. Subsequent works have proposed several refinements to address these issues. InstaSplat (Fan et al., 2024) uses DUSt3R (Wang et al., 2024b), a transformer-based dense stereo model, to initialize dense point clouds from sparse-view images. However, it primarily focuses on stereo matching, resulting in less accurate camera parameters. In contrast, VGGSfM (Wang et al., 2024a) introduces a fully differentiable SfM pipeline with deep learning integration at every stage, providing highly accurate camera poses beneficial for downstream tasks, including 3D reconstruction (Heo, 2024). Additionally, 3DGS employs a suboptimal, heuristic-based densification strategy for growing Gaussians. To improve this, various works (Bulò et al., 2024; Zhang et al., 2024; Ye et al., 2024) have focused on redesigning error criteria to densify Gaussians. Notably, Kheradmand et al. (2024) reformulates Gaussian updates as state transitions in MCMC samples, stabilizing training dynamics by recomputing updated opacity and scale values when cloning Gaussians.

## 3 METHOD

Given an SCI measurement capturing a 3D scene and modulation masks, we aim to reconstruct the target 3D scene and render high-quality encoded images in real-time. We first propose a novel initialization protocol to estimate point clouds and poses from a single SCI measurement to start the training procedure. Then, we jointly optimize Gaussians and poses by minimizing photometric loss between the synthesized and real measurements. Besides, We also leverage the MCMC strategy (Kheradmand et al., 2024) to stabilize the training process.

An overview of the proposed pipeline is illustrated in Figure 2. This section first briefly reviews the background knowledge of 3DGS and the image formation model of video SCI (Section 3.1). Then

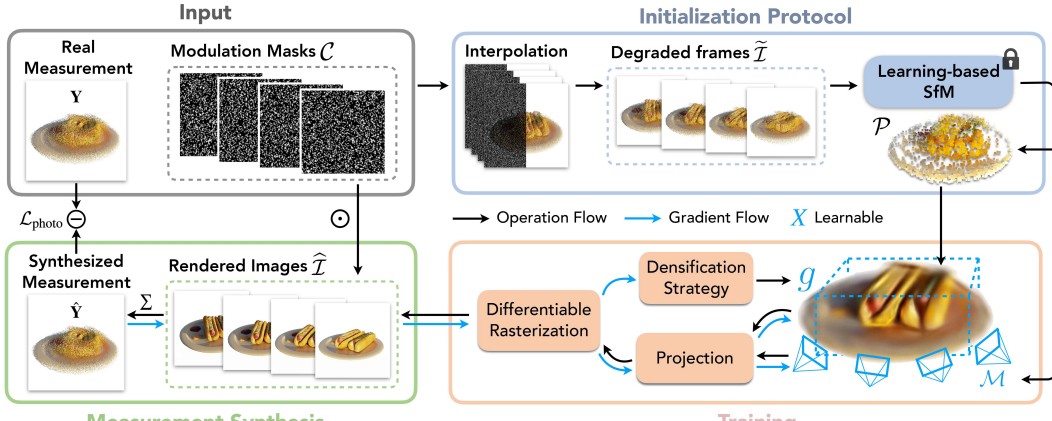

Figure 2: **Overview of the Proposed Pipeline.** SCISplat utilizes 3DGS to reconstruct 3D scene from a single SCI image. The process begins with the initialization protocol, where a set of degraded frames $\widetilde{\mathcal{I}}$ is extracted from the real measurement $\mathbf{Y}$ and modulation masks $\mathcal{C}$ using interpolation. These frames are then input into a learning-based SfM module to generate rough estimates of the point cloud $\mathcal{P}$ and camera poses $\mathcal{M}$, which are the initial parameters for the 3D Gaussians $g$. SCISplat then performs joint optimization of the Gaussians and their corresponding camera poses. Adhering to the image formation model of SCI, it transfers the rendered images $\widehat{\mathcal{I}}$ from differentiable Gaussian rasterization to produce a synthesized measurement $\hat{\mathbf{Y}}$. The optimization process minimizes the photometric loss $\mathcal{L}_{\text{photo}}$ between the synthesized and real SCI measurements while also incorporating regularization losses for the opacity and scale of the Gaussians, $\mathcal{L}_{\text{opacity}}$ and $\mathcal{L}_{\text{scale}}$, respectively.

Section 3.2 discusses how to initialize point cloud and camera poses from an SCI measurement. Next, we introduce the forward process of the pipeline, which combines 3DGS rendering and the SCI image formation model. Finally, we provide insights into our choice of MCMC training strategy to compensate for the ill-posed photometric loss computed with SCI measurements (Section 3.3).

### 3.1 PRELIMINARIES

#### 3.1.1 BACKGROUND ON 3DGS

A set of 3D Gaussians $g = \{\mathbf{g}_i\}_{i=1}^{M}$, parameterized by mean position $\boldsymbol{\mu}_i \in \mathbb{R}^3$, 3D covariance $\boldsymbol{\Sigma}_i \in \mathbb{R}^{3\times3}$, opacity $\mathbf{o}_i \in \mathbb{R}$ and color $\mathbf{c}_i \in \mathbb{R}^3$, are introduced to faithfully represent the 3D scene. The input contains a series of multi-view images $\mathcal{I} = \{\mathbf{I}_i \in \mathbb{R}^{H\times W}\}_{i=1}^{N}$ of the target 3D scene as well as their corresponding projection matrices $\mathcal{M} = \{\mathbf{M}_i \in \mathbb{R}^{3\times4}\}_{i=1}^{N}$ and point cloud of the scene. Then 3DGS renders multi-view images $\widehat{\mathcal{I}} = \{\widehat{\mathbf{I}}_i \in \mathbb{R}^{H\times W}\}_{i=1}^{N}$ posed at $\mathcal{M}$ through the differentiable Gaussian rasterization

$$\widehat{\mathcal{I}} = R(g, \mathcal{M}) \tag{1}$$

where $R(\cdot, \cdot)$ represent the rendering function that renders images posed at $\mathcal{M}$ with existing 3D Gaussians $g$. Then existing Gaussians $g$ are updated by minimizing the photometric loss computed between rendered images $\widehat{\mathcal{I}}$ and real captured images $\mathcal{I}$.

$$g^* = \arg\min_{g} \mathcal{L}_{\text{photo}}(\widehat{\mathcal{I}}, \mathcal{I})$$
$$\mathcal{L}_{\text{photo}} = (1 - \lambda_{\text{D-SSIM}}) \cdot \mathcal{L}_1 + \lambda_{\text{D-SSIM}} \cdot \mathcal{L}_{\text{D-SSIM}} \tag{2}$$

where $\mathcal{L}_1$ is the average L1 distance, and $\mathcal{L}_{\text{D-SSIM}}$ is the Structural Similarity Index Metric (SSIM) between the rendered $\widehat{\mathcal{I}}$ and ground-truth image $\mathcal{I}$. As in Zhao et al. (2017), $\lambda_{\text{D-SSIM}}$ is set to 0.2.

#### 3.1.2 IMAGE FORMATION MODEL OF VIDEO SCI

The formation process of a video SCI system is similar to that of a motion-blurred image. The difference is that the captured images $\mathcal{I} = \{\mathbf{I}_i \in \mathbb{R}^{H\times W}\}_{i=1}^{N_I}$ is modulated by $N_I$ binary masks $\mathcal{C} = \{\mathbf{C}_i \in \mathbb{R}^{H\times W}\}_{i=1}^{N_I}$ across exposure time, where $H$ and $W$ are image height and width respectively. Those masks are achieved by displaying different 2D patterns on the Digital Micro-mirror Device (DMD) and a spatial light modulator. The image sensor then accumulates the modulated photons

across exposure time to a compressed image $\mathbf{Y} \in \mathbb{R}^{H \times W}$. The number of masks or different patterns on the DMD within exposure time determines the number of coded frames, i.e., the temporal compression ratio (CR). This whole encoding process can be formally stated as follows:

$$\mathbf{Y} = \sum_{i=1}^{N_I} \mathbf{I}_i \odot \mathbf{C}_i + \mathbf{Z}, \tag{3}$$

where $N_I$ is the temporal CR, $\odot$ denotes element-wise multiplication, and $\mathbf{Z} \in \mathbb{R}^{H \times W}$ is the measurement noise. The individual pixel value in the binary mask $\mathbf{C}_i$ is randomly generated. For $N_I$ masks across exposure time, the probability of assigning 1 to the same pixel location is fixed.

## 3.2 INITIALIZATION PROTOCOL FROM AN SCI MEASUREMENT

Different from prior NeRF-based methods, to initiate the training of 3DGS, an initial point cloud that serves as a coarse approximation of the 3D scene is needed. It brings challenges with SCI measurement since it is usually too noisy to be used by COLMAP Schönberger & Frahm (2016). In addition, camera poses within exposure time are also required to render images to form the synthesized measurement. To overcome those challenges, we propose a novel initialization protocol for SCISplat.

Inspired by Wang et al. (2023), we first normalize the real measurement $\mathbf{Y}$ by the sum of all modulation masks $\mathbf{C}_i$

$$\overline{\mathbf{Y}} = \mathbf{Y} \oslash \sum_{i=1}^{N} \mathbf{C_i}, \tag{4}$$

where $\overline{\mathbf{Y}}$ is the normalized measurement, and $\oslash$ denotes element-wise division. Then, degraded frames $\widetilde{\mathcal{I}} = \{\widetilde{\mathbf{I}}_i \in \mathbb{R}^{H \times W}\}_{i=1}^{N_I}$ can be obtained by interpolating the normalized measurement after modulated by a filtered version of each mask $\mathbf{C}_i \odot \mathbf{B}_i$,

$$\widetilde{\mathbf{I}}_i = \text{Interp}\left(\overline{\mathbf{Y}} \odot (\mathbf{C}_i \odot \mathbf{B}_i)\right), \quad (\mathbf{B}_i)_{j,k} = \begin{cases} 1, & \text{if } (\mathbf{C}_i)_{j,k} >= \tau \\ 0, & \text{otherwise} \end{cases}, \tag{5}$$

where $\mathbf{B}_i$ is a selection matrix that only preserve the value of $\mathbf{C}_i$ positioned at $(j, k)$ if its value exceeds $\tau$. For synthetic data $\tau = 1$, since modulation masks only contain 0 and 1. For real data, $\tau$ is carefully selected as 0.8 to filter out measurement noise, whose effects are shown in Figure 5.

As visualized in Figure 2, large parts of these degraded frames $\widetilde{\mathcal{I}}$ are contaminated by noise, which makes traditional feature tracking approaches fail easily. Therefore, COLMAP can not be directly utilized here like most 3DGS-based pipelines. However, with the recent development of learning-based SfM methods, obtaining a decent guess from these noisy images is possible. Specifically, we use VGGSfM (Wang et al., 2024a) denoted as $f_\theta(\cdot)$, a fully differentiable SfM pipeline, to directly get an initial point cloud $\mathcal{P} = \{\mathbf{P}_i \in \mathbb{R}^3\}_{i=1}^{N_P}$ and camera projection matrix estimates $\mathcal{M} = \{\mathbf{M}_i \in \mathbb{R}^{3 \times 4}\}_{i=1}^{N_I}$ from degraded frames $\widetilde{\mathcal{I}}$, whose deep point tracker is relatively robust to noise presented in degraded frames.

$$\mathcal{P}, \mathcal{M} = f_\theta(\widetilde{\mathcal{I}}). \tag{6}$$

This initial point cloud $\mathcal{P}$ will inevitably contain noisy points. Still, it would not affect the reconstruction quality much after downsampling since 3DGS only requires a coarse approximation of the 3D scene at the start of training. It is noteworthy that each projection matrix $\mathbf{M}_i$ consists of extrinsics $\mathbf{T}_i \in \mathbb{SE}(3)$, which define camera poses, and intrinsics $\mathbf{K} \in \mathbb{R}^{3 \times 3}$. The $\mathbf{T}_i$ will be optimized, but $\mathbf{K}$ will remain fixed for all frames throughout training.

## 3.3 TRAINING SCHEME

**Gaussians Initialization.** Given that our point cloud $\mathcal{P}$ is directly derived from noisy degraded frames $\widetilde{\mathcal{I}}$, it will indeed contain some noisy points that will harm the rendering quality. Also, having many points at the start of training will quickly introduce artifacts due to inaccurate poses $\mathcal{M}$. Thus, we uniformly downsample the initial point cloud $\mathcal{P}$ to a certain number $n$ as $\mathcal{P}_n$ to mitigate the formation of noisy artifacts at the early stage. On the other hand, having too few points also challenges

the densification strategy's ability to faithfully reconstruct the scene, given that we are recovering the scene from a compressed measurement. Therefore, choosing a suitable number of initial points is vital to the final reconstruction quality, further investigated in Table 5. After downsampling, the initial set of $n$ Gaussians $g = \{\mathbf{g}_i\}_{i=1}^n$ are placed at the location of these subsampled points $\mathcal{P}_n$.

**Optimization Strategy and Loss Function.** At each iteration, images $\widehat{\mathcal{I}}$ posed at $\mathcal{M}$ are rendered with current set of Gaussians $g$ following Equation 1. Then following Equation 3, a synthesized measurement is formed by modulating the rendered images $\widehat{\mathcal{I}}$ with corresponding masks $\mathcal{C}$ as shown below:

$$\widehat{\mathbf{Y}} = \sum_{i=1}^{N_I} \widehat{\mathbf{I}}_i \odot \mathbf{C}_i, \tag{7}$$

where $\widehat{\mathbf{Y}} \in \mathbb{R}^{H \times W}$ represents the synthesized measurement. Here, we emulate the image formation process of video SCI and omit the measurement noise term $\mathbf{Z}$ in Equation 3 to facilitate the recovery of originally captured images $\mathcal{I}$. We then compute the photometric loss between the synthesized output $\widehat{\mathbf{Y}}$ and the real measurement $\mathbf{Y}$, which forms a crucial component of the loss function:

$$\mathcal{L}_{\text{total}} = \mathcal{L}_{\text{photo}}(\widehat{\mathbf{Y}}, \mathbf{Y}) + \lambda_{\text{opacity}} \cdot \mathcal{L}_{\text{opacity}}(g) + \lambda_{\text{scale}} \cdot \mathcal{L}_{\text{scale}}(g). \tag{8}$$

Here, the latter two terms are adapted from Kheradmand et al. (2024), which minimize the scale and opacity of the current Gaussians $g$ to encourage a lower number of effective Gaussians. With this loss, we jointly optimize Gaussians $g$ and camera poses $\mathcal{M}$ as

$$g^*, \mathcal{M}^* = \arg\min_{g, \mathcal{M}} \mathcal{L}_{\text{total}}. \tag{9}$$

By optimizing the camera poses $\mathcal{M}$ alongside the Gaussians $g$, we can compensate for inaccuracies in the initial poses estimated from degraded frames, thereby enhancing the reconstruction quality.

**Densification Strategy.** Regarding the densification strategy for Gaussians, we employ MCMC strategy (Kheradmand et al., 2024) instead of the original for the reasons below. In the original strategy, the composed opacity values are larger after cloning or splitting Gaussians, making the scene appear slightly brighter. This inconsistent update introduces spiky appearance changes in the scene, which sometimes can make camera poses drift to sub-optimal locations due to the unstable gradient flow from photometric loss, thus collapsing the whole reconstruction. Moreover, due to pixel ambiguity in fitting an SCI measurement, suddenly having high opacity for Gaussians will quickly introduce noise in the reconstructed viewpoint. Specifically, we optimize Gaussians by summing pixels of rendered images, as indicated by Equation 7, to fit the SCI measurement. With this ill-posed loss, a sub-optimal solution could quickly occur where one pixel has a high pixel value close to the sum while others appear much darker. In this case, that pixel will appear as noise on the reconstructed images. Recalling the original densification strategy, suddenly making individual Gaussian opacity high will undoubtedly encourage the reach of this local optimal and thus lead to noisy reconstruction. In contrast, the MCMC strategy corrects this opacity bias by recomputing the updated opacity values after densification, resulting in smoother training dynamics and effective noise suppression.

## 4 EXPERIMENTS

We validate our SCISplat on synthetic and real datasets and evaluate it against existing state-of-the-art SCI image restoration and 3D reconstruction methods. The experimental results demonstrate that SCISplat delivers higher performance in image restoration quality and achieves significantly faster training and rendering speed.

### 4.1 EXPERIMENTAL SETUP

**Dataset.** To ensure a fair and all-rounded comparison, we test our method on the same synthetic and real datasets as SCINeRF (Li et al., 2024). The synthetic dataset contains six scenes with different resolutions: 512×512, 400×400, and 600×400, which shows the effectiveness of our method under different resolutions. The real dataset has a higher resolution of 1024×768, further challenging the ability of our method to recover high-resolution authentic images.

Table 1: **Quantitative results on SCI image reconstruction task with synthetic datasets.** The first part shows the performance of conventional SCI methods, and then their reconstructed images are used to train 3DGS which forms the second part of the table. At the bottom are the results of pure 3D methods that incorporate the SCI image formation process in the forward pass, highlighting that our method delivers the best performance across the board. Colors denote the 1st , 2nd and 3rd best performing model.

| Method | Airplants | | | Hotdog | | | Cozy2room | | | Tanabata | | | Factory | | | Vender | | |
|---|---|---|---|---|---|---|---|---|---|---|---|---|---|---|---|---|---|---|
| | PSNR↑ | SSIM↑ | LPIPS↓ | PSNR↑ | SSIM↑ | LPIPS↓ | PSNR↑ | SSIM↑ | LPIPS↓ | PSNR↑ | SSIM↑ | LPIPS↓ | PSNR↑ | SSIM↑ | LPIPS↓ | PSNR↑ | SSIM↑ | LPIPS↓ |
| GAP-TV | 22.85 | 0.406 | 0.499 | 22.35 | 0.766 | 0.318 | 21.77 | 0.432 | 0.603 | 20.42 | 0.426 | 0.625 | 24.05 | 0.566 | 0.515 | 20.00 | 0.368 | 0.688 |
| PnP-FFDNet | 27.79 | 0.912 | 0.182 | 29.00 | 0.977 | 0.051 | 28.98 | 0.892 | 0.984 | 29.17 | 0.903 | 0.119 | 31.75 | 0.897 | 0.114 | 28.70 | 0.923 | 0.131 |
| PnP-FastDVDNet | 28.18 | 0.909 | 0.175 | 29.93 | 0.972 | 0.052 | 30.19 | 0.913 | 0.079 | 29.73 | 0.933 | 0.098 | 32.53 | 0.916 | 0.105 | 29.68 | 0.939 | 0.104 |
| EfficientSCI | 30.13 | 0.942 | 0.112 | 30.75 | 0.956 | 0.046 | 31.47 | 0.932 | 0.047 | 32.30 | 0.958 | 0.060 | 32.87 | 0.925 | 0.070 | 33.17 | 0.940 | 0.045 |
| GAP-TV+3DGS | 24.08 | 0.430 | 0.493 | 24.03 | 0.794 | 0.233 | 22.81 | 0.535 | 0.399 | 22.35 | 0.523 | 0.410 | 26.40 | 0.717 | 0.390 | 22.20 | 0.476 | 0.436 |
| PnP-FFDNet+3DGS | 28.51 | 0.917 | 0.194 | 30.22 | 0.979 | 0.074 | 31.01 | 0.915 | 0.092 | 32.67 | 0.947 | 0.100 | 31.76 | 0.925 | 0.112 | 31.98 | 0.954 | 0.109 |
| PnP-FastDVDNet+3DGS | 28.71 | 0.911 | 0.152 | 30.62 | 0.980 | 0.069 | 31.48 | 0.916 | 0.093 | 33.47 | 0.953 | 0.092 | 32.09 | 0.935 | 0.095 | 32.93 | 0.958 | 0.103 |
| EfficientSCI+3DGS | 30.32 | 0.943 | 0.115 | 31.79 | 0.924 | 0.049 | 32.26 | 0.934 | 0.059 | 33.73 | 0.968 | 0.057 | 34.06 | 0.955 | 0.080 | 33.31 | 0.975 | 0.049 |
| SCINeRF | 30.69 | 0.933 | 0.072 | 31.35 | 0.987 | 0.031 | 33.23 | 0.949 | 0.044 | 33.61 | 0.963 | 0.037 | 36.60 | 0.963 | 0.022 | 36.40 | 0.984 | 0.029 |
| Ours | 31.45 | 0.951 | 0.036 | 32.67 | 0.991 | 0.016 | 35.26 | 0.972 | 0.011 | 37.86 | 0.985 | 0.005 | 38.92 | 0.975 | 0.010 | 39.49 | 0.992 | 0.004 |

Table 2: **Quantitative comparisons of training time (hrs) and inference/rendering speed (FPS) of different methods on the synthetic dataset.** Our method prevails on both training time and inference speed.

| Method | GAP-TV | PnP-FFDNet | PnP-FastDVDNet | EfficientSCI | SCINeRF | Ours |
|---|---|---|---|---|---|---|
| Training Time (hrs) ↓ | N/A | N/A | N/A | $\approx$100 hrs | $\approx$10 hrs | <1 hrs |
| Inference Speed (FPS) ↑ | 0.13 | 0.01 | 0.01 | 2.6 | 0.25 | 205 |

**Baseline Methods and Evaluation Metrics.** Since SCISplat can render high-quality images from estimated 3D scenes, we compare our method against SOTA SCI image/video reconstruction frameworks, including model-based methods such as GAP-TV (Yuan, 2016), and deep-learning-based methods including PnP-FFDNet (Yuan et al., 2020), PnP-FastDVDNet (Yuan et al., 2021b), EfficientSCI (Wang et al., 2023) and SCINeRF (Li et al., 2024). For fair comparisons, we fine-tuned EiffcientSCI using masks from synthetic datasets and trained SCINeRF from scratch for each scene in synthetic/real datasets.

For image synthesis quality, we employ commonly-used metrics, including structural similarity index (SSIM), peak signal-to-noise ratio (PSNR), and learned perceptual image patch similarity (LPIPS) (Zhang et al., 2018). Since our SCISplat can optimize camera poses, we also evaluated the camera pose estimation capabilities of SCISplat by computing the absolute translation error (ATE), a widely used camera pose estimation evaluation metric in visual odometry.

**Implementation Details.** We implemented our method using PyTorch (Paszke et al., 2019) within the 3DGS (Kerbl et al., 2023) pipeline. We leveraged Adam optimizer (Kingma & Ba, 2014) to optimize Gaussians and camera poses. The learning rate for Gaussians is scaled by the square root of 8 since we are forwarding 8 images at once according to the square root rule (André et al., 2022). The learning rate for camera poses is exponentially decreased from $5 \times 10^{-4}$ to $2.5 \times 10^{-7}$. We set the maximum number of Gaussians as 100000 for MCMC (Kheradmand et al., 2024) strategy. All experiments are conducted on an NVIDIA RTX 4090 GPU.

## 4.2 RESULTS

The experimental results from the synthetic dataset provide robust empirical evidence regarding the efficacy of SCISplat in estimating and representing 3D scenes from a single SCI image, as demonstrated in Figure 3 and Table 1. Our method demonstrates superior performance compared to SOTA SCI image reconstruction algorithms and SCINeRF with an improvement of 2.3 dB in terms of PSNR. Additionally, we evaluated SCISplat against naive SCI 3D scene representation approaches by utilizing reconstructed 2D images from SOTA SCI methods in conjunction with conventional 3DGS models. We employed camera poses and point clouds derived from ground truth images to initialize these naive baselines. Although it introduces an unfair comparison to our method, SCISplat still consistently outperforms these naive two-stage approaches. It demonstrates the necessity to model SCI image formation model within the pipeline.

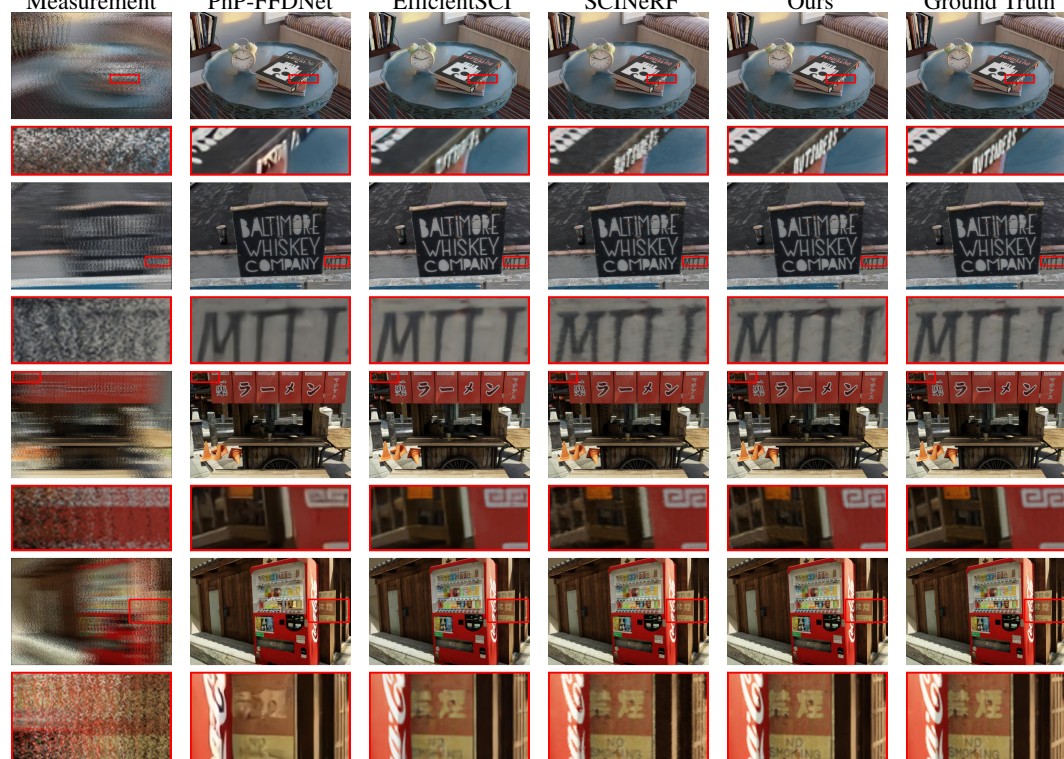

Figure 3: **Qualitative evaluations of our method against SOTA SCI image restoration methods on the synthetic dataset.** Top to bottom shows the results for different scenes, including *Cozy2room*, *Factory*, *Tanabata*, and *Vender*. The experimental results demonstrate that our method achieves superior performance on image restoration from a single SCI image (the far-left column).

Table 3: **Pose estimation performance of SCINeRF and SCISplat on synthetic dataset.** The results are in the ATE metric. SCINeRF enforces camera poses lie in a linear trajectory to achieve lower error in scenes with straight-line motion (e.g. *Cozy2room*, *Tanabata* and *Vender*), while our method optimizes individual camera poses without such a constraint, which fits better to complex trajectories.

| ATE↓ | Airplants | Hotdog | Cozy2room | Tanabata | Factory | Vender |
|---|---|---|---|---|---|---|
| SCINeRF | 0.00502 ± 0.00171 | 0.02068±0.00696 | 0.00015 ± 0.00008 | 0.00016±0.00007 | 0.00065 ± 0.00048 | 0.00023 ± 0.00008 |
| Ours | 0.00459 ± 0.00189 | 0.01536±0.00439 | 0.02028 ± 0.01296 | 0.05117 ± 0.02411 | 0.00059 ± 0.00043 | 0.00231 ± 0.00118 |

Furthermore, we assessed the computational efficiency of various SCI reconstruction algorithms as shown in Table 2. Conventional SCI reconstruction methods require extensive training, which takes hours or even days, and they cannot achieve real-time inference. In contrast, our SCISplat completes training within 1 hour and achieves >200 FPS for image inference and rendering, which are 10× (train) and 820× (inference) faster than SCINeRF.

For pose optimization performance, Table 3 shows the absolute translation error that SCINeRF and SCISplat achieve on each scene, respectively. Notably, the ATE of our SCISplat on some scenes in the synthetic dataset (*Cozy2room, Tanabata, Factory*) are larger than SCINeRF. This phenomenon can be attributed to the fact that ground truth camera trajectories in these scenes are straight lines, which perfectly fits the linear trajectory assumption of SCINeRF. On the other hand, our SCISplat performs better on scenes with more complicated camera trajectories, such as *Airplants* and *Hotdog* with curvature trajectories.

To evaluate the performance of SCISplat on real datasets, we conduct qualitative comparisons against SOTA methods. Figure 4 presents the experimental results, illustrating outcomes for real datasets. Notably, existing SOTA methods for SCI image reconstruction show limitations in handling high-frequency details and characters, leading to significant deficiencies in output images. Although SCINeRF exhibits improved generalization capabilities by accurately recovering these details, it introduces additional "granular" noise in rendered images. In contrast, SCISplat outperforms

| Measurement | PnP-FFDNet | EfficientSCI | SCINeRF | Ours | Original Scene |
| --- | --- | --- | --- | --- | --- |

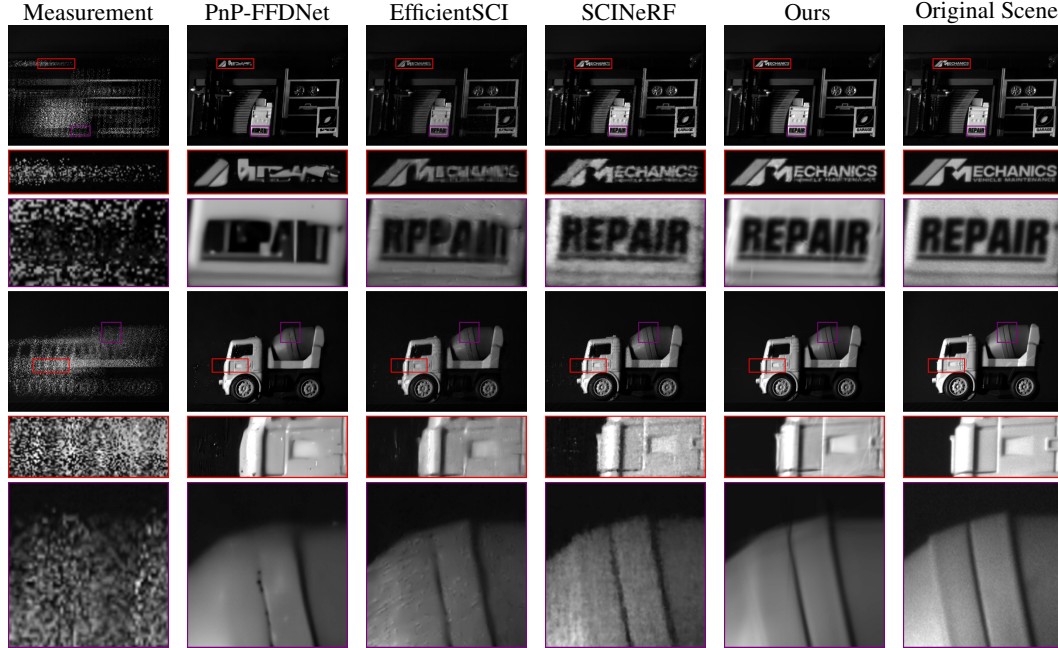

Figure 4: **Qualitative evaluations of our method against SOTA SCI image restoration methods with real dataset captured by SCI system.** Top to bottom shows the results for different scenes, Since the pixel-wise aligned ground truth images in real datasets are unavailable, we capture separate scene images after capturing the SCI image for reference. The experimental results show that our SCISplat surpasses existing image restoration methods by recovering intricate details and outperforms SCINeRF by avoiding high-frequency noises.

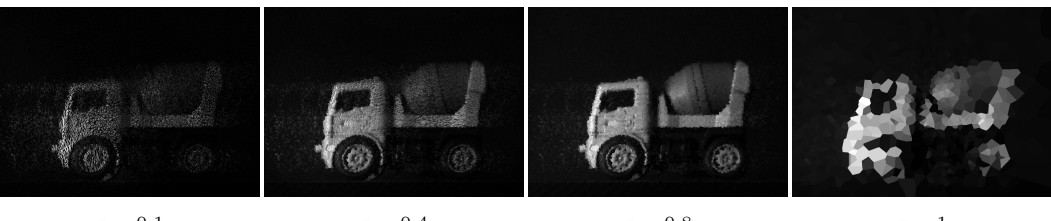

| $\tau = 0.1$ | $\tau = 0.4$ | $\tau = 0.8$ | $\tau = 1$ |
| --- | --- | --- | --- |

Figure 5: **Ablation studies on different thresholds $\tau$ for interpolating real data.** At a low threshold $\tau = 0.1$, the interpolated image is overwhelmed by measurement noise. At a high threshold $\tau = 1$, the image retains only a vague outline of the captured object. Therefore, we select $\tau = 0.8$ as a balanced compromise.

these methods on real datasets by effectively recovering scenes with fine details while eliminating the high-frequency noise presented in SCINeRF.

## 4.3 ABLATION STUDY

**Interpolation Threshold.** As shown in Figure 5, varying the interpolation threshold results in a trade-off between noise reduction and detail preservation. We select a threshold of 0.8 to recover degraded frames from real datasets, as it maintains sufficient detail for the SfM module to operate effectively without introducing excessive noise.

**SfM methods used for Initialization.** As shown in Table 4, we conduct experiments on synthetic datasets with different SfM methods to initialize point cloud and camera poses. For COLMAP, it fails to find valid matches with degraded frames and thus cannot be used in our protocol. DUSt3R gives dense point clouds, but inaccurate camera poses degrade the reconstruction quality. VGGSfM, in contrast, gives reliable point cloud and camera poses, thus leading to the best results.

**Number of Initial Points.** To assess the impact of our down-sampling operation on final rendering quality, we selected two scenes with a high number of initial points: 15519 for *Airplants* and 28249

Table 4: **Ablation studies on different SfM methods used in initialization.** COLMAP fails to find any matches with degraded frames and thus cannot be used in our initialization protocol. DUSt3R gives dense points but inaccurate poses, which degrades the performance. VGGSfM consistently gives reliable point cloud and camera poses, reflected by the final reconstruction quality.

| Initialization | PSNR↑ | SSIM↑ | LPIPS↓ |
|---|---|---|---|
| COLMAP | × | × | × |
| DUSt3R | 32.12 | 0.9408 | 0.0524 |
| VGGSfM | 35.94 | 0.9777 | 0.0143 |

Table 5: **Ablation Studies on Different Downsample Thresholds for Initial Points.** *"w/o downsample"* indicates that no downsampling is applied to the initial point cloud. Subsequently, the downsample threshold $n$ is set to 10000, 5000, and 1000 to determine the optimal value.

| Threshold $n$ | Airplants | | | Factory | | |
|---|---|---|---|---|---|---|
| | PSNR↑ | SSIM↑ | LPIPS↓ | PSNR↑ | SSIM↑ | LPIPS↓ |
| *w/o* downsample | 30.56 | 0.9374 | 0.0599 | 38.92 | 0.9755 | 0.0099 |
| 10000 points | 31.45 | 0.9507 | 0.0364 | 38.92 | 0.9755 | 0.0100 |
| 5000 points | 31.36 | 0.9481 | 0.0401 | 38.51 | 0.9741 | 0.0104 |
| 1000 points | 30.91 | 0.9444 | 0.0420 | 37.95 | 0.9697 | 0.0129 |

Table 6: **Ablation studies on densification strategy and pose optimization.** This figure shows the average metrics on synthetic datasets, demonstrating the impact of enabling or disabling pose optimization. The results underscore the importance of using the MCMC strategy with pose optimization to achieve optimal performance.

| Strategy | w/ pose optimization | | | w/o pose optimization | | |
|---|---|---|---|---|---|---|
| | PSNR↑ | SSIM↑ | LPIPS↓ | PSNR↑ | SSIM↑ | LPIPS↓ |
| Original | 33.73 | 0.9654 | 0.0295 | 32.54 | 0.9563 | 0.0423 |
| MCMC | 35.94 | 0.9777 | 0.0143 | 34.43 | 0.9719 | 0.0137 |

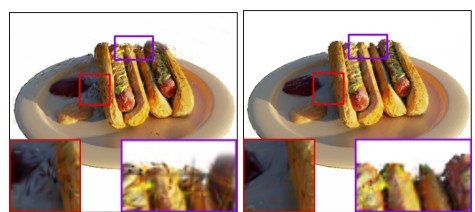

Original      MCMC

Figure 6: **Qualitative results with different densification strategies.** The original densification strategy introduces high opacity noise around the edges.

for *Factory*. These scenes represent different trajectories, with *Airplants* having a curved trajectory and *Factory* exhibiting a linear trajectory. In Table 5, we evaluate our model by training it without downsampling and by downsampling to 10000, 5000, and 1000 points, respectively. This allows us to observe how the number of initial points influences rendering quality. The results indicate that training with the full set of points, which likely contains noise, degrades rendering quality, especially in the *Airplants* scene. In contrast, the *Factory* scene exhibits relatively minor camera motions, resulting in less noise in the degraded frames. Consequently, this leads to a cleaner point cloud that may not require downsampling. Conversely, insufficient initial points negatively impact performance in both scenes. This is due to the densification strategy's inability to grow Gaussians effectively from limited coverage.

**Densification Strategy.** As shown in Table 6, employing the MCMC strategy vastly improves the average metrics on rendering quality. Also observed in Figure 6, sticking with the original densification strategy forms more floaters and high-opacity artifacts around the edges.

**Pose Optimization.** As shown in Table 6, "*w/o* pose optimization" disables the refinement of camera poses while optimizing Gaussians, which leads to significant degradation in rendering quality. This is because the initial poses estimated from noisy degraded frames could be sub-optimal.

## 5 CONCLUSION

In this paper, we introduce SCISplat, the first 3DGS-based approach for efficiently reconstructing a 3D scene from a single snapshot compressive image. SCISplat leverages 3D Gaussian Splatting as its core scene representation, offering remarkable scene fidelity and exceptional speed in both training and rendering. To initiate the training of SCISplat, we propose a novel initialization protocol that robustly derives the initial point cloud and camera poses from an SCI measurement. By incorporating the SCI image formation model into the 3DGS training pipeline, our method achieves real-time rendering of high-quality scene images from a single SCI measurement. Extensive experimental results demonstrate that SCISplat outperforms all state-of-the-art methods in terms of rendering quality, training efficiency, and inference speed on both synthetic and real datasets.

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

# A    APPENDIX

In the appendix, we present complementary qualitative results on the synthetic dataset for the un-covered two scenes (*Airplants*, *Hotdog*). We also run the naive two-stage baseline for SCINeRF and provide the quantitative results.

## A.1    COMPLEMENTARY RESULTS ON SYNTHETIC DATASET

This section presents the qualitative comparisons between our SCISplat and SOTA methods on two synthetic scenes uncovered by the main paper: *Airplants* and *Hotdog*, as shown in Figure 4. The qualitative results illustrate that our SCISplat can recover intricate details without introducing high-frequency noises in SCINeRF.

| Measurement | PnP-FFDNet | EfficientSCI | SCINeRF | Ours | Original Scene |

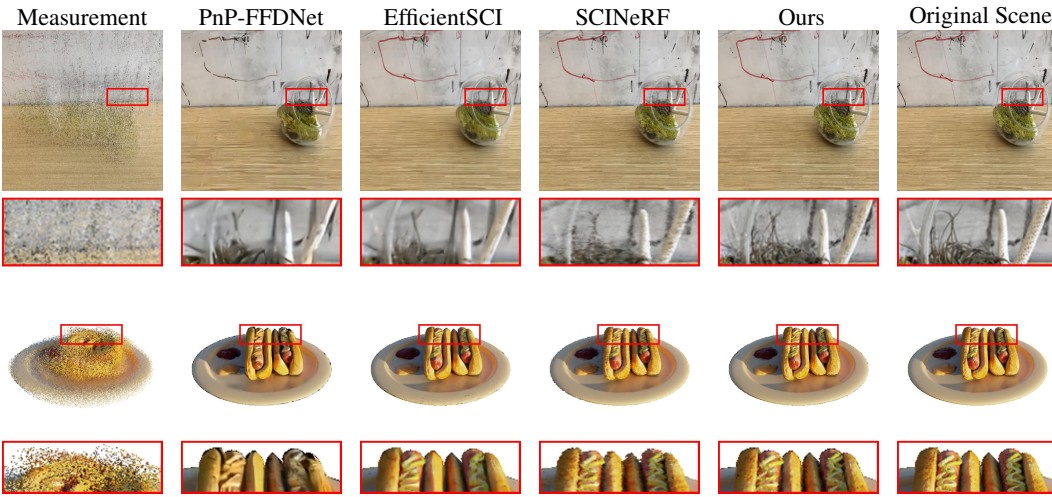

Figure 7: **Qualitative evaluations of our method against SOTA SCI image restoration methods on the synthetic dataset.** Top to bottom shows the results for two scenes in the synthetic dataset that are not covered by the main paper: *Airplants* and *Hotdog*. The experimental results demonstrate that our method achieves superior performance on image restoration from a single SCI image (the far-left column).

## A.2    QUANTITATIVE RESULTS ON NAIVE BASELINES OF NERF

Table 7: **Quantitative results of naive NeRF baselines on SCI image reconstruction task with synthetic datasets.** Our method outperforms all naive baselines of NeRF across the board. Colors denote the `1st`, `2nd` and `3rd` best performing model.

| Method | Airplants | | | Hotdog | | | Cozy2room | | | Tanabata | | | Factory | | | Vender | | |
|---|---|---|---|---|---|---|---|---|---|---|---|---|---|---|---|---|---|---|
| | PSNR↑ | SSIM↑ | LPIPS↓ | PSNR↑ | SSIM↑ | LPIPS↓ | PSNR↑ | SSIM↑ | LPIPS↓ | PSNR↑ | SSIM↑ | LPIPS↓ | PSNR↑ | SSIM↑ | LPIPS↓ | PSNR↑ | SSIM↑ | LPIPS↓ |
| NeRF+GAP-TV | 23.29 | 0.407 | 0.549 | 22.10 | 0.777 | 0.292 | 21.91 | 0.481 | 0.568 | 21.19 | 0.489 | 0.534 | 25.09 | 0.667 | 0.439 | 28.94 | 0.807 | 0.568 |
| NeRF+PnP-FFDNet | 27.38 | 0.891 | 0.216 | 26.93 | 0.957 | 0.068 | 29.99 | 0.908 | 0.084 | 29.99 | 0.956 | 0.094 | 31.45 | 0.920 | 0.103 | 30.02 | 0.946 | 0.104 |
| NeRF+PnP-FastDVDNet | 27.65 | 0.888 | 0.164 | 27.33 | 0.959 | 0.064 | 30.30 | 0.908 | 0.086 | 32.25 | 0.948 | 0.087 | 31.87 | 0.931 | 0.099 | 31.70 | 0.955 | 0.089 |
| NeRF+EfficientSCI | 29.13 | 0.910 | 0.139 | 31.63 | 0.928 | 0.050 | 32.03 | 0.929 | 0.058 | 33.91 | 0.966 | 0.060 | 33.33 | 0.950 | 0.077 | 32.88 | 0.975 | 0.051 |
| SCINeRF | 30.69 | 0.933 | 0.072 | 31.35 | 0.987 | 0.031 | 33.23 | 0.949 | 0.044 | 33.61 | 0.963 | 0.037 | 36.60 | 0.963 | 0.022 | 36.40 | 0.984 | 0.029 |
| Ours | 31.45 | 0.951 | 0.036 | 32.67 | 0.991 | 0.016 | 35.26 | 0.972 | 0.011 | 37.86 | 0.985 | 0.005 | 38.92 | 0.975 | 0.010 | 39.49 | 0.992 | 0.004 |

