# OpenReview forum: "SCISplat: 3D Gaussian Splatting from a Snapshot Compressive Image"
_ICLR.cc/2025/Conference — ICLR 2025 Conference Withdrawn Submission_

### Official Review · Reviewer_8Rjt · 2024-10-29

**Soundness:** 3
**Presentation:** 3
**Contribution:** 2
**Rating:** 5
**Confidence:** 4

**Summary:**

This paper restores the SCI (Snapshot Compressive Images) with 3DGS. With the benefit of 3DGS, it can achieve high-quality and fast rendering, compared to NeRF-based approaches. Degraded frames are used to acquire initial points, along with a learning-based SfM method. Moreover, the Monte Carlo Markov Chain serves as a densification strategy, aiding in an abrupt change in opacity, which will cause rendering noises. With all these designs, this paper achieves the best among compared baselines, both in rendering quality and speed.

**Strengths:**

1. The purpose of this article is highly meaningful. SCI imaging, as a method of information encoding, can significantly reduce imaging costs. However, using traditional decoding methods may result in the loss of multi-view information. Therefore, I believe introducing NeRF or 3DGS as auxiliary restoration methods is quite reasonable.
2. The experiment section is very detailed; I especially appreciate the ablation study part, where each module is thoroughly examined, including the SfM method and the number of initial points.

**Weaknesses:**

1. Although the task itself is quite meaningful, I didn’t see many particularly innovative contributions in this article. Compared with SCINeRF, I feel the core contribution is modifying NeRF into 3DGS, which achieves higher-quality rendering and faster training and rendering speeds.
2. Let’s return to the initialization stage. Firstly, the initialization itself is introduced by 3DGS, so it’s hard to attribute this module as a novel contribution—it’s more of a necessary design when utilizing 3DGS. Secondly, this part of the design isn’t irreplaceable; at the very least, I believe that images obtained using traditional SCI recovery methods could be used for SfM calibration, even if their 3D consistency isn’t perfect, as the precision required for the initial point cloud is not very high. Therefore, I may not admit the initialization process is indispensable.
3. The densification process (MCMC) is also not the author's original idea and, therefore, cannot be considered their contribution.

**Questions:**

1. Perhaps the authors could present the average scores of their method compared to other baselines. This would provide a more intuitive understanding and make it easier to compare with the metrics used in the ablation study.

---

### Official Review · Reviewer_jUA7 · 2024-10-31

**Soundness:** 3
**Presentation:** 3
**Contribution:** 2
**Rating:** 5
**Confidence:** 4

**Summary:**

To address the challenge of high-quality and rapid reconstruction of compressed measurements images, this paper proposes a Gaussian Splatting (3DGS) initialization protocol based on the 3DGS framework. This protocol enables the acquisition of point clouds and poses from compressed measurements as the inputs of 3DGS, thereby achieving the decoding of SCI. It has demonstrated clearer reconstruction quality and faster reconstruction speeds on both synthetic and real-world data.

**Strengths:**

+ The paper has a good background introduction and literature presentation on video SCI and 3DGS techniques.

+ The SCIplat proposed in this paper shows better reconstruction quality across different datasets and enables faster training and rendering.

+ The SCIplat proposed in this paper outperforms the two-stage approach, arguing for the need to jointly optimize the camera pose and the Gaussian function.

**Weaknesses:**

- In order to obtain good reconstruction quality, the article describes quite a few measures such as initialization protocols, point cloud down-sampling, MCMC strategy, and loss functions, but these all seem to come from existing techniques.

- As shown in Table 6, the MCMC densification strategy greatly improves the metrics of this paper, but unfortunately, it seems difficult to see the authors' technological innovation there.

- Considering that this paper has a much faster training speed and inference/rendering speed, we seem to think that this paper has a stronger engineering value.

**Questions:**

- Equation 5 represents the process of obtaining degenerate frames from normalized measurements, where interp() has any specific meaning. Please describe it in detail in conjunction with Fig. 2.

- As shown in Table 3, the proposed SCIplat performs slightly better than the SOTA method when processing scenes with complex camera trajectories. However, when processing scenes with simple linear trajectories, there is a two-order-of-magnitude gap between SCIplat's metrics and the compared methods. We expect the authors to provide more experimental results for scenes with different trajectory complexities or detailed theoretical analyses to further illustrate the mechanism of this paper's method in handling different types of trajectories.

- Combining Fig 3 and 4, the proposed method performs more prominently on real datasets than synthetic data, especially in comparison with the SCINeRF method. It is recommended to supplement the ablation experiments on synthetic and real datasets concerning the framework or the threshold value in order to verify the relative importance of the impact of framework choices and threshold settings on performance.

- As shown in Table 5, different down-sampling has different effects on different scenarios, how to choose a more general sampling strategy for more scenarios?

---

### Official Review · Reviewer_Mrvr · 2024-11-02

**Soundness:** 4
**Presentation:** 3
**Contribution:** 2
**Rating:** 5
**Confidence:** 5

**Summary:**

Following previous work using NeRFs (SCINeRF), the paper proposes a method to reconstruct 3D scenes from a single temporally compressed image using 3D Gaussian Splatting instead. In order to do so, the authors design an initialization with a point cloud for the initial 3D Gaussian means and camera poses estimated by an existing learning-based SfM pipeline (VGGSfM) that is robust enough to be applied to degraded frames decoded from the Snapshot Compressive Imaging (SCI) measurement. After initialization, both Gaussians and camera poses are optimized jointly to fit the SCI image. An evaluation on synthetic and real data shows that the proposed method outperforms state-of-the-art baselines in image reconstruction quality while being fast to train and render.

**Strengths:**

- The paper tackles an interesting and challenging problem of reconstructing 3D scenes from compressed data with possible applications in memory- or bandwidth-critical scenarios.
- The evaluation provides strong empirical results validating the method's effectiveness in terms of both
  - quality (state-of-the-art image reconstruction on synthetic datasets with large gaps to the best baseline SCINeRF [1], better details and less noisy qualitative results for real scenes) and
  - efficiency (much faster training and rendering compared to SCINeRF [1]).
- Extensive ablation studies across hyperparameters (interpolation threshold, number of initial points) and adopted components from existing works (SfM method, densification strategy) validate the design choices.
- The paper is well written and easy to understand.

References:
[1] SCINeRF: Neural Radiance Fields from a Snapshot Compressive Image. CVPR 2024

**Weaknesses:**

- Lack of novelty:
  - Incremental: Compared to the main baseline SCINeRF [1], the idea is to effectively replace NeRF with 3D Gaussian Splatting [2] to benefit from its high quality and fast training and rendering.
   - Training and rendering speed as one of the main advantages over previous work is only due to the use of 3DGS instead of NeRF.
  - The main technical contribution is the "initialization protocol" consisting of two steps:
    1.  a simple decoding consisting of masking and interpolation to obtain "degraded frames"
    2. application of an existing learning-based SfM approach VGGSfM [3], which turns out to be sufficiently robust to output an appropriate initialization point cloud and set of camera poses.
  - All major remaining ideas / components have been adopted from previous work:
    - Photometric SCI loss from [3]
    - Joint optimization of camera poses and 3D representation like in [3]
    - Densification strategy from [4]
- Some lack of clarity regarding novelty:
  - The paper lacks some clarity regarding the novelty of leveraging the SCI image in the loss for optimization, which is adopted from SCINeRF [1]:
    - The authors write:
      - "we apply a specifically designed loss function by incorporating the SCI image formuation model with the 3DGS training procedure" (lines 74ff.)
      - "we emulate the image formation process of video SCI and ..." (lines 281 f.)
    - The whole section 3.3 is missing a reference to SCINeRF [1], which introduced this loss originally.
  - Similarly, it is not very clear from the paper that the previous work [1] also already optimizes the camera poses / trajectory jointly with the 3D representation. Besides mentioning it in the related work, a reference to [1] in the description of that part of the proposed method would help avoiding misunderstandings.
- Limited evaluation setup:
  - The evaluation is limited to image reconstruction only, while the main baseline SCINeRF [1] additionally evaluates novel view synthesis results on views not included in the SCI measurement.
  - Similar to [1], a comparison across different compression rates would be interesting.
  - The evaluation on real data is limited to qualitative comparisons only. If possible, quantitative results would be helpful.
- Some lack of contextualization w.r.t. baselines (please see questions):
  - It is unclear how flexible previous generalizable approaches are compared to per scene optimization approaches (SCINeRF [1] and the proposed method) w.r.t.:
    - static vs dynamic scenes
    - camera trajectories and scene coverage
  - The comparison of training time with EfficientSCI [5] as one example of a deep learning-based approach seems to be unfair, as the proposed method has to be optimized per 3D scene, whereas this class of baselines should generalize to unseen scenes after training.

Minor comment:
- In section 3.1.2, the modulation masks are said to be binary, while in section 3.2 thresholding is used to binarize them. This is a bit confusing.

References:
- [1] SCINeRF: Neural Radiance Fields from a Snapshot Compressive Image. CVPR 2024
- [2] 3D Gaussian Splatting for Real-Time Radiance Field Rendering. SIGGRAPH 2023
- [3] VGGSfM: Visual Geometry Grounded Deep Structure From Motion. CVPR 2024
- [4] 3D Gaussian Splatting as Markov Chain Monte Carlo. NeurIPS 2024
- [5] EfficientSCI: Densely Connected Network with Space-time Factorization for Large-scale Video Snapshot Compressive Imaging. CVPR 2023

**Questions:**

- Is there any reason why quantitative comparisons on the real dataset would not be possible?
- Does the proposed method struggle compared to generalizable approaches for dynamic scenes or certain camera trajectories?

---

### Official Review · Reviewer_dzVX · 2024-11-03

**Soundness:** 3
**Presentation:** 2
**Contribution:** 2
**Rating:** 5
**Confidence:** 2

**Summary:**

This paper introduces SCISplat, a novel method for 3D scene reconstruction from a single snapshot compressive image (SCI). SCISplat is the first approach leveraging 3D Gaussian Splatting (3DGS) to reconstruct 3D scene structure from a single SCI, addressing challenges in point cloud and camera pose initialization, as well as joint Gaussian and camera pose optimization. Experimental results demonstrate that SCISplat outperforms previous methods, achieving superior reconstruction quality and efficiency.

**Strengths:**

1. The first SCI decoding method based on 3D Generative Synthesis (3DGS) achieves high time efficiency and superior quality.
2. Comprehensive ablation studies demonstrate the effectiveness of the proposed components.
3. Achieve sota performance on both synthetic and real datasets.

**Weaknesses:**

1. The paper presents limited novelty, as the initialization uses the learned structure-from-motion method VGGSfM, optimization follows ordinary objective functions, and densification processes with MCMC are primarily derived from existing papers. There is not much inspiration that the combination and adaptation of existing techniques specifically address the challenges of SCI data.
2. Lack of clarity; for instance, the methodology employed for optimizing camera poses during the training process is not adequately explained.

**Questions:**

1. As weakness 2, could you provide a more detailed explanation of the camera pose optimization process? The experimental results indicate the performance of pose estimation, but the specifics of how these poses are optimized during training remain unclear.
2. Given that the learned method (VGGSfm) offers improved initialization of camera poses, how do you anticipate other methods, such as SCINeRF, would perform if initialized with the same parameters? Would there be a notable difference in their results?
3. To my understanding, 3DGS can effectively operate with randomly initialized point clouds. Is this also applicable to SCI data? Additionally, could you provide some visualizations demonstrating the initialization process of VGGSfM?

---

### Comment · Area_Chair_xWpZ · 2024-11-20

Dear reviewers and authors, this is a reminder that November 13 to November 26 at 11:59pm AoE: Reviewers and authors can exchange responses with each other as often as they wish. Thanks!

---

### Note · Authors · 2024-11-22

I have read and agree with the venue's withdrawal policy on behalf of myself and my co-authors.